# Entity-Based Evaluation of Political Bias in Automatic Summarization

**Karen Zhou**
The University of Chicago
karenzhou@uchicago.edu

**Chenhao Tan**
The University of Chicago
chenhao@uchicago.edu

## Abstract

Growing literature has shown that NLP systems may encode social biases; however, the *political* bias of summarization models remains relatively unknown. In this work, we use an entity replacement method to investigate the portrayal of politicians in automatically generated summaries of news articles. We develop an entity-based computational framework to assess the sensitivities of several extractive and abstractive summarizers to the politicians Donald Trump and Joe Biden. We find consistent differences in these summaries upon entity replacement, such as reduced emphasis of Trump's presence in the context of the same article and a more individualistic representation of Trump with respect to the collective US government (i.e., administration). These summary dissimilarities are most prominent when the entity is heavily featured in the source article. Our characterization provides a foundation for future studies of bias in summarization and for normative discussions on the ideal qualities of automatic summaries.

## 1 Introduction

Automatic summarization aims to identify the *most important* information in input documents and produce a concise summary to save readers time and effort (Nenkova and McKeown, 2011). However, the pursuit of efficiency may come with undesired biases, especially in a political context. Bias of a political nature can contribute to issues like polarization and misinformation. It is also subjective and difficult to tell on the basis of a single article.

In this work, we conduct the first empirical study to examine the political bias of automatic summarizers based on large-scale pre-trained language models. We investigate the portrayal of politicians in automatically generated summaries through an entity replacement experiment design; the advantage of this setup is the ability to control for the content of the source document. Figure 1 shows

an example: given a news article about Joe Biden, we replace all occurrences of Biden's name with that of Donald Trump and examine the differences between the summary of the original article and that of the replaced article.

What should an ideal summarization model output for these two versions of an article, *which differ only by one entity name*? This normative question is crucial to consider for responsible deployment and usage of summarization models. A first step towards addressing this query is to uncover whether state-of-the-art automatic summarizers actually encode any bias at all. In this paper, we focus on this empirical demonstration and leave the deliberation of desirable differences for future work (see §5).

We demonstrate substantial differences across four summarization models. The consistent biases in the generated summaries suggest robustly different model representations of the two politicians. Furthermore, *entity-centric* news articles, those that heavily feature the original entity, lead to more dissimilar summaries upon replacement. **We acknowledge that "bias" is an overloaded term;** in this paper, we use it to refer to **significant differences with respect to entity representation.** We do not endorse the interpretation that the summaries are "anti-Trump/anti-conservative".

In summary, our main contributions are:

- We conduct the first study on the political bias in automatic summarization.

- We propose an entity replacement approach to compare, model representations of key political figures.

- We find consistent differences, such as under-emphasis of Trump's presence with the same article context and more individualistic representation of Trump in relation to the administration. These variations reveal that summarization models are not neutral with respect to political entities.

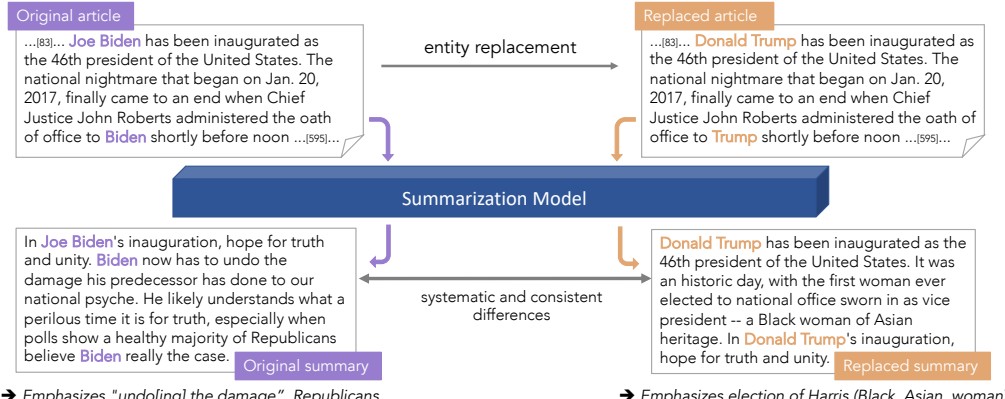

**Figure 1**: Illustration of our experimental design. Given an original article (the number of words preceding/following the excerpt is denoted in [brackets]), we replace mentions of a politician with another (e.g., Biden → Trump). This example shows that the model generates different summaries upon entity replacement in the source article.

## 2 Related Work

There are two primary types of automatic summaries: *extractive* and *abstractive* (El-Kassas et al., 2021). Neural summarization models have demonstrated superior performance in benchmark datasets (e.g., Zhang et al. (2020)). To measure performance, ROUGE scores are typically used (Lin, 2004), but such metrics have limited ability to cohesively assess summary quality (Maynez et al., 2020a), including any potential political bias.

There is strong resemblance between selecting the *most important* information in the summarization process and the phenomenon of media framing, where media outlets selectively report aspects of an issue or quotes of politicians (Entman, 1993; Chong and Druckman, 2007). The selection process by media outlets has received substantial attention in computational research (Niculae et al., 2015; Tan et al., 2018; Guo et al., 2022; Vallejo et al., 2023), Our identified problem differs in that the content selection is done by models rather than media outlets and that the kind of bias may not align with the conservative vs. liberal biases as defined for media framing.

A growing line of work investigates the biases of NLP systems to understand their potential harms if/when deployed (Sheng et al., 2021). Many studies have covered representation bias and social bias (e.g., gender and race) (Steen and Markert, 2023; Sheng et al., 2019; Liang et al., 2021; Yeo and Chen, 2020; Garimella et al., 2021; Dhamala et al., 2021). In comparison, the harm of political bias is more subtle; it can be difficult to observe in an individual article, but can still have substantial impact (e.g., affecting elections in US swing states). While Feng et al. (2023) tracks political bias in language

models for hate speech and misinformation detection, our work is the first study to reveal political bias in automatic summarization models.

In addition, prior work has examined factual consistency in generated texts (Kryscinski et al., 2020; Devaraj et al., 2022; Huang et al., 2021; Maynez et al., 2020a; Tang et al., 2022), fairness of extractive summaries with classic summarization methods (Shandilya et al., 2018), and generating neutral summaries to mitigate media bias (Lee et al., 2022). Nan et al. (2021) take an entity-based approach to evaluating factuality in abstractive summaries.

Replacement paradigms similar to our own have been used to uncover biases in other NLP applications, such as language modeling (Feder et al., 2021; Vig et al., 2020; Patel and Pavlick, 2021) and neural machine translation (Wang et al., 2022).

## 3 Methodology

In this section, we introduce our data, experimental design, and the models we evaluate. Data and code are available at https://github.com/ChicagoHAI/entity-based-political-bias.

**Data:** For *source documents*, we use cleanly-parsed articles from the NOW corpus[1] from 1/1/2020 to 12/31/2021 that have at most 5,000 words. Among these news articles that mention any US politician, about 62.9% include Donald Trump and/or Joe Biden. Thus, we focus our investigation of bias on the entities Trump and Biden: well-known political figures that play important roles in US news. We separate the articles into two experimental groups: 157,648 articles that mention

---

[1] https://www.english-corpora.org/now/.

| | |
|---|---|
| $p < 1e\text{-}20$: | $\uparrow\uparrow\uparrow\uparrow$ / $\downarrow\downarrow\downarrow\downarrow$ |
| $p < 0.001$: | $\uparrow\uparrow\uparrow$ / $\downarrow\downarrow\downarrow$ |
| $p < 0.01$: | $\uparrow\uparrow$ / $\downarrow\downarrow$ |
| $p < 0.05$: | $\uparrow$ / $\downarrow$ |
| $p \geq 0.05$: | — / — |

Table 1: Two-sided $t$-test thresholds and notations. The direction of the arrows indicates which class has the larger mean; i.e., $\uparrow$ indicates that $S_i^{e_2}$ has the larger mean, while $\downarrow$ corresponds to a larger mean in $S_i^{e_1}$.

Trump only and 85,952 articles that mention Biden only.[2] Appendix A contains additional statistics.

**Replace and compare:** Our experimental design consists of two steps. Given a pair of politicians of interest $(e_1, e_2)$, we replace all the occurrences of $e_1$ in each news article with $e_2$ ($e_1 \rightarrow e_2$ replacement). We then have $N$ paired documents where the content only differs by mentions of these politicians: $\{(D_i^{e_1}, D_i^{e_2})_{i=1}^N\}$. For a model $m$, we input each news article to $m$ and obtain summaries $\{(S_i^{e_1}, S_i^{e_2})_{i=1}^N\}_m$.

We substitute each $e_1 \in \{\text{Trump, Biden}\}$ with each $e_2 \in \{\text{Trump, Biden, Obama, Bush}\} \setminus \{e_1\}$ in this paper. Presidents Barack Obama and George W. Bush are included as $e_2$ candidates to verify the robustness of our findings. Appendix A provides extra details about these instantiations.

The second step is to assess the differences between these two sets of corresponding summaries $S_i^{e_1}$ and $S_i^{e_2}$. We define feature functions to extract properties of these summaries, and then conduct paired $t$-tests to measure significance levels. From the patterns arising in the results of these analyses, we develop an understanding of the models' encoded biases for representations of the entities.

**Models:** The summarization models used in this work include an extractive model and three abstractive models. Specifically, we use the following:

- PRESUMM (Liu and Lapata, 2019): BERT-based model with hierarchical encoders, designed and fine-tuned for extractive text summarization.

- PEGASUS (Zhang et al., 2020): model with Gap Sentence Generation objective, pre-trained specifically for the abstractive summarization task. Training data $\sim 46 \times$ PRESUMM's.

- BART (Lewis et al., 2019): model with denoising corrupted input document objective,

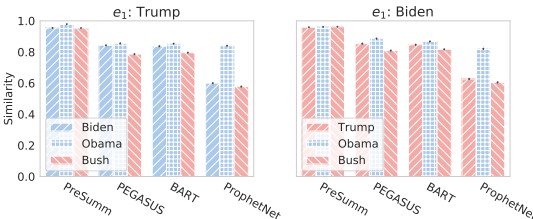

Figure 2: PRESUMM summaries have the highest similarity ratios, while PROPHETNET summaries are most dissimilar after entity replacement.

pre-trained for general NLG tasks. Training data $\sim 10 \times$ PRESUMM's.

- PROPHETNET (Qi et al., 2020): model with future n-gram prediction (using n-stream self-attention) objective, pre-trained for general sequence-to-sequence tasks. Training data $\sim 10 \times$ PRESUMM's.

All three abstractive models are Transformer-based encoder-decoders that are among the best performing with respect to ROUGE scores. They are also all fine-tuned for summarization on the CNN/DailyMail dataset (Hermann et al., 2015), consisting of news from 2007-2015. We use the default settings for all generation models as this is likely the most common use case in applications.

## 4 Framework and Results

We present our results for similarity between summaries, and two feature functions: 1) entity name mentions in summaries and 2) temporal differences in status indicators. We focus on these two features in this short paper to zero in on the basic notion of representation in sensitivity to political entities.

All results involve comparing frequencies via paired $t$-tests; we present results using the notation described in Table 1. The numerical mean differences and $p$-values can be obtained from our publicly released code (see §3). As described in Appendix B, there are small but signifcant differences in summary lengths before and after replacement. Therefore, all frequencies are normalized by length of summary before the $t$-tests.

### 4.1 Overall Similarity of $S_i^{e_1}$ and $S_i^{e_2}$

We first compare the *similarity scores* between the summary pairs.[3] Figure 2 shows that PRESUMM

---

[2]During the $e_1 \rightarrow e_2$ entity replacement step, we consider only the articles that mention $e_1$ but not $e_2$.

[3]We use SequenceMatcher from difflib, which defines this score as the ratio $2.0 * M/T$, where $T$ is the total number of words in both sequences, and $M$ is the number of matches, based on the Gestalt approach (Ratcliff and Metzener, 1988).

| $e_1$ | Trump | | | Biden | | |
|---|---|---|---|---|---|---|
| $e_2$ | Biden | Obama | Bush | Trump | Obama | Bush |
| PRESUMM | ↑↑↑ | ↓↓↓↓ | ↑↑ | — | ↓↓↓↓ | ↓↓↓ |
| PEGASUS | ↑↑↑↑ | ↑↑↑↑ | ↑↑↑↑ | ↓↓↓↓ | ↓↓↓↓ | ↓↓↓↓ |
| BART | ↑↑↑↑ | ↑↑↑↑ | ↑↑↑↑ | ↓↓↓↓ | ↓↓↓ | ↑↑↑↑ |
| PROPHETNET | ↑↑↑↑ | ↑↑↑↑ | ↑↑↑↑ | ↓↓↓↓ | ↑↑↑↑ | ↑↑↑↑ |

Table 2: Trump is less likely to be mentioned in abstractive summaries than the other presidents, suggesting that these models exhibit inclusion bias. The consistent ↑ arrows for Trump → $e_2$ show that replacing Trump's name with $e_2$ in the source article results in abstractive summaries with significantly higher frequencies of $e_2$; this observation appears consistent for Biden → Trump.

summaries have the highest similarity ratios as expected, since the summaries are directly extracted from the articles and the dissimilarity only stems from differences in selection. PROPHETNET summaries have the lowest similarities; this most powerful and latest summarization model before GPT-3 (see Limitations) is the most sensitive to differences in entity names, suggesting that it may encode the most differences between the two politicians. Appendix B contains additional comparisons.

When investigating what factors are correlated with dissimilar summaries after entity replacement, we find that **entity-centric articles lead to more dissimilar summaries.** From the Trump → Biden and Biden → Trump scores, we split the news articles into ones leading to highly similar summaries (HIGHSIM) vs. those with highly dissimilar summaries (LOWSIM) for each model.[4] After creating each model's HIGHSIM and LOWSIM article subsets, we identified distinguishing key words with the Fightin' Words algorithm with the uninformative Dirichlet prior (Monroe et al., 2009). We observe that the greater presence of $e_1$'s name consistently leads to dissimilar summaries for the $e_1 \rightarrow e_2$ directions. This indicates that *entity-centric* articles lead to dissimilar summaries upon replacement; i.e., the model's representational differences emerge as Trump or Biden becomes the focus of the article. See Appendix B for detailed results from this exploration.

## 4.2 Entity Representation in Summaries

We then determine whether the summary pairs mention their entities to the same extent. This property reflects *inclusion bias* (Steen and Markert, 2023): the idea that entities should be mentioned in a summary with equal frequency if they are equally salient in the source document. We define this feature function as the frequency with which the entity is named in the summary, i.e., $\#e_1$ in summary, and likewise for $e_2$ after replacement.[5]

**Trump is least likely to be mentioned in abstractive summaries.** The abstractive summarizers tend *not* to mention Trump's name (see Table 2). There are also significant differences in usage of Biden's name compared to the other presidents. This result suggests that such summaries are sensitive to the entities in the article. In particular, they de-emphasize Trump's presence compared to other presidents in the context of the same article, which may skew reader perception of the event. See Appendix C for example summary pairs.

## 4.3 Indicators of Status Over Time

An important event during the time period of our data is the inauguration of Biden at the start of 2021, which changes his status from former Vice President to President. To identify changes across time due to this event, we separate the articles by year of publication (2020 vs. 2021), and then assess the frequency of the title "Vice President" across generated summaries $S_i^{e_1}$ and $S_i^{e_2}$.

Another relevant change is the use of the word "administration" (e.g., "the Biden administration"), which can be seen as an indicator of whether the presidency is viewed as an individualistic or a collective effort. Sunstein and Lessig (1994) also draw distinctions between the president and the administration, and their perceived authority. The frequency of this term is another feature function.

**Most models associate Biden with being the Vice President.** From the frequency of "Vice President" usage in 2020 vs. 2021 (Table 3), Biden is more strongly associated with the title of "Vice President". This observation holds even in 2021 summaries, when he assumes the title "President". Our observation is aligned with recent work on hallucination and factuality (Ji et al., 2022; Maynez et al., 2020b), highlighting their potential impacts in political news summarization.

**Abstractive models tend not to associate Trump with the "administration".** The abstractive models consistently dissociate Trump with the use of "administration" (Table 4). This result suggests that

---

[4]We define these thresholds using the 75th- and 25th-percentiles respectively.

[5]To ensure fair comparison, we restore $e_1$ in $S_i^{e_2}$ where it makes sense.

| $e_1$ | Trump | | | | | | Biden | | | | | |
|---|---|---|---|---|---|---|---|---|---|---|---|---|
| $e_2$ | Biden | | Obama | | Bush | | Trump | | Obama | | Bush | |
| | 20 | 21 | 20 | 21 | 20 | 21 | 20 | 21 | 20 | 21 | 20 | 21 |
| PRESUMM | ↑ | ↑↑ | — | — | — | — | ↓↓↓ | — | ↓↓↓ | ↓↓↓ | ↓ | ↓ |
| PEGASUS | ↑↑↑↑ | ↑↑↑↑ | — | — | ↓ | — | ↓↓↓ | ↓↓↓ | ↓↓↓ | ↓↓↓ | ↓ | ↓↓↓ |
| BART | ↑↑↑ | ↑↑↑ | ↑↑↑ | — | ↑↑↑ | — | ↓↓ | ↓↓ | ↓↓↓ | ↓↓ | ↑ | — |
| PROPHETNET | ↑↑↑ | ↑↑↑ | — | — | — | — | ↓↓↓ | ↓ | ↓↓↓ | ↓↓↓ | ↓↓ | ↓↓↓ |

Table 3: Biden is more likely to be associated with "Vice President" in nearly all cases. The consistent ↓ arrows for Biden → $e_2$ show that replacing Biden's name with $e_2$ in the source article results in summaries with significantly lower inclusion frequencies of "Vice President"; the phenomenon is consistent for Trump → Biden.

| $e_1$ | Trump | | | | | | Biden | | | | | |
|---|---|---|---|---|---|---|---|---|---|---|---|---|
| $e_2$ | Biden | | Obama | | Bush | | Trump | | Obama | | Bush | |
| | 20 | 21 | 20 | 21 | 20 | 21 | 20 | 21 | 20 | 21 | 20 | 21 |
| PRESUMM | ↓↓↓ | ↓↓↓ | — | ↓ | ↑ | — | — | ↑↑↑↑ | — | ↑↑↑ | — | ↑↑↑↑ |
| PEGASUS | ↑↑↑↑ | ↑↑↑ | ↑↑↑↑ | ↓↓ | ↑↑↑↑ | — | — | ↓↓↓↓ | ↓↓ | ↓↓↓↓ | ↓↓↓ | ↓↓↓ |
| BART | ↑↑↑↑ | ↑↑ | ↑↑↑↑ | ↑↑↑ | ↑↑↑↑ | ↑↑↑ | — | ↓↓↓↓ | — | ↓ | — | ↑↑↑ |
| PROPHETNET | ↑↑↑↑ | ↑↑↑ | ↑↑↑↑ | ↑↑↑ | ↑↑↑↑ | — | — | ↓↓↓↓ | ↓↓ | — | — | ↓↓↓↓ |

Table 4: Trump is less likely to appear with the word "administration" in all abstractive summaries, suggesting that those models hold a less unified view of him with respect to the US government than the other presidents. The consistent ↑ arrows for Trump → $e_2$ show that replacing Trump's name with $e_2$ in the source article results in summaries with significantly more mentions of "administration"; this observation is consistent for Biden → Trump.

Trump's actions may not be representative of those of the broader US government; in other words, these models encode a more individualistic representation of Trump compared to other presidents. Examples of these results are in Appendix C.

## 5 Conclusion

In this work, we conduct the first empirical study on political bias in summarization models. Through experiments that replace the names of Donald Trump and Joe Biden with other presidential entities in US news articles, we identify several areas of variation in summaries when the source article differs by a single person's name. We show that **summarization models learn different representations for different entities and provide consistently different summaries depending on the entities involved**. Our framework can be used for further analysis of biases in summarization models.

Politicians are often in similar situations, such as signing a bill or meeting a world leader. Models with distinct representations for different entities will generate different summaries for such reports. These variations challenge the assumption that summaries are inherently neutral, that one can produce or seek a single gold-standard automatic summary. Such differences and their downstream effects (e.g., on voter preferences) warrant further exploration.

Ultimately, we must consider: **what constitutes an ideal automatic summary?** This is an important normative query, which we purposefully do not answer in this paper. Having distinct representations of politicians may indicate a nuanced model, but that does not automatically justify all differences. Characterizing the systematic biases encoded in automatic summarizers is a first step towards responsible use of such models. Our work aims to motivate discussion of this essential yet overlooked research area.

## Limitations

We acknowledge the limitations of our methodology, which provide opportunities for further research. This proposed framework was only evaluated on US politicians, specifically two US presidents. There are limits to studying only these two entities, and not other less popular/controversial politicians. We attempt to account for this by adding the additional $e_2$ candidates Obama and Bush. Additionally, because Trump and Biden are prominent political figures that feature heavily in US political news (see §3), the consistent biases we found across multiple summarization models constitute important initial findings. Summaries of such high-profile politicians are more likely to be read and have possible harms. Furthermore, different from the emphasis on generalization in modeling work, scholars in other fields such as political science may dedicate an entire book to the study of a single politician (e.g., Wilson (2016)).

Political bias may vary across different countries and regimes. We also do not claim that the observed differences are exclusive to the two politicians that we study here. They likely exist for any notable entities in the models' training data.

Another limitation lies in our work's exclusive focus on English news articles. While the frequency metrics can generalize to other nations and languages, some analyses done here may be specific to English terms and dynamics of US politics. We also do not consider co-reference resolution of politician titles in entity replacement. Entity-swapping between politicians of different genders may require additional data processing due to the need for pronoun resolution.

Our work uses the default generation setup in the four models of interest. We recognize that there exist a multitude of hyperparameter settings and decoding algorithms that can lead to summaries of different properties. Future studies can also examine the impact of decoding algorithms on biases in automatic summarization.

Finally, while GPT-3+ has been shown to perform well on some summarization tasks (Goyal et al., 2022; Yang et al., 2023; Zhang et al., 2023), we do not include it since our study began prior to the API's accessibility. Furthermore, the models we evaluated are publicly available, which allows for greater control over the generation of summaries. As political bias is a sensitive social issue with subjective connotations, we believe that it is important to conduct a fair and thorough comparison in a controllable setting. Our proposed framework can be readily applied to GPT-3+ and other models in future evaluations.

## Ethics Statement

Having demonstrated systematic biases related to entity replacement, one could knowingly manipulate a summary of a news article about Trump by replacing his name with Biden's, and then restoring his original name in the adversarial summary. This action is related to spinned summaries (Bagdasaryan and Shmatikov, 2021), and there may be concerns about contributing to fake news and disinformation (Buchanan et al., 2021).

On the other hand, this research contributes to revealing potential biases and harms of NLP systems. A primary benefit of our findings is improving public understanding of NLP models, which may lead to cautious deployment of such models. We believe that this is especially important for summarization, which is often portrayed as a straightforward task. Ultimately, we believe that the benefits of being able to characterize political biases in automatic summarization will aid in developing defenses to such adversarial use cases. Raising awareness of such biases can also enable an informed discussion on what desirable summaries should include and how summarization models should be used.

Overall, our work adds to the emerging area of understanding potential issues of NLP systems, and it makes an important contribution towards recognizing the political biases of summarization models, which is critical for their responsible use.

## Acknowledgements

We thank our anonymous reviewers for their helpful comments. We also thank Chenghao Yang, Shi Feng, and the other members of the Chicago Human+AI lab for their feedback and support. This work was in part supported by the AI & democracy research initiative at the University of Chicago. Karen Zhou is additionally supported by a GFSD Fellowship.

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

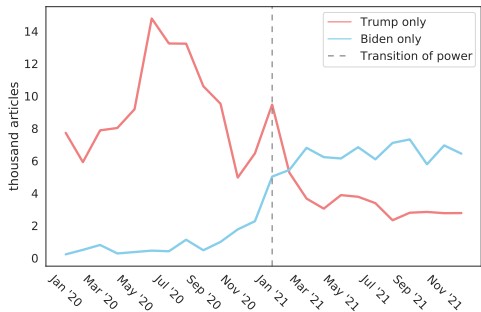

Figure 3: Number of articles by month. The quantity of "Trump only" articles declines after the inauguration in January 2021, while "Biden only" articles increase.

| Trump | ↔ | Biden |
|---|---|---|
| Donald Trump, D. Trump | ↔ | Joe Biden, Joseph Biden, J. Biden |
| Donald J. Trump, Donald John Trump | ↔ | Joe R. Biden, Joseph R. Biden, Joe Robinette Biden, Joseph Robinette Biden |

Table 5: Mapping between names for the entity replacement. If the name form has multiple mappings (e.g., "Donald Trump" → {"Joe Biden", "Joseph Biden"}), the first name in the list is used as the default value.

Xianjun Yang, Yan Li, Xinlu Zhang, Haifeng Chen, and Wei Cheng. 2023. Exploring the limits of chatgpt for query or aspect-based text summarization. *ArXiv*, abs/2302.08081.

C. Yeo and A. Chen. 2020. Defining and Evaluating Fair Natural Language Generation. *WINLP*.

Haopeng Zhang, Xiao Liu, and Jiawei Zhang. 2023. Extractive summarization via chatgpt for faithful summary generation. *ArXiv*, abs/2304.04193.

Jingqing Zhang, Yao Zhao, Mohammad Saleh, and Peter J. Liu. 2020. PEGASUS: Pre-training with Extracted Gap-sentences for Abstractive Summarization. *arXiv:1912.08777 [cs]*.

## A Methodology Details

**Data statistics:** The "Trump only" articles average 744.8 words in length and the "Biden only" articles average 808.0 words in length. Figure 3 shows the distribution of articles that include these entity mentions. "Trump only" articles decrease in number after the inauguration in January 2021, while the number of "Biden only" articles increases.

As stated in §3, among 2020-2021 news articles that mention any US politician, about 62.9% include Donald Trump and/or Joe Biden. We filter the articles mentioning any US politician as ones that include names from CivilServiceUSA. Then, among those articles, we search for the fraction that also mention "Trump" and "Biden".

| $e_1$ | Trump | | | Biden | | |
|---|---|---|---|---|---|---|
| $e_2$ | Biden | Obama | Bush | Trump | Obama | Bush |
| PRESUMM | ↑↑↑ | ↓↓↓ | ↑↑↑↑ | ↓↓↓ | ↓↓↓ | ↑↑↑ |
| PEGASUS | ↓↓↓↓ | ↓↓↓↓ | ↓↓↓↓ | ↑↑↑↑ | ↓↓ | ↓↓↓ |
| BART | ↓↓↓ | ↓↓↓ | ↓↓↓↓ | ↑↑↑ | — | ↓↓↓↓ |
| PROPHETNET | ↓↓↓↓ | ↑↑↑↑ | ↑↑↑↑ | — | ↑↑↑↑ | ↑↑↑↑ |

Table 6: Summary length $t$-test results. Abstractive summaries tend to be longer for Trump.

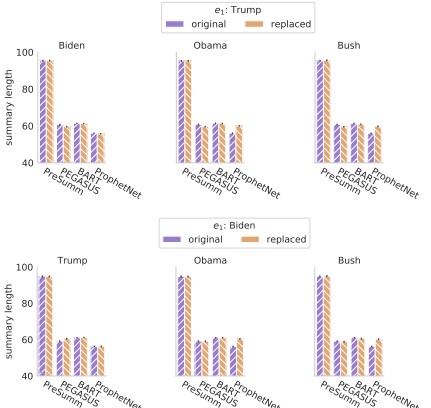

Figure 4: There are small but significant differences in length between the original and replaced summaries.

**Additional replace and compare details:** To do the $e_1 \rightarrow e_2$ replacement, we create mappings between the different variations of the entity names. In the case of Trump and Biden, we use the name mapping in Table 5 to do the Trump $\rightarrow$ Biden and Biden $\rightarrow$ Trump replacements. Similar mappings are utilized for Obama and Bush.

## B  Additional Summary Similarity Comparisons

**Comparing lengths and similarity scores of $S_i^{e_1}$ and $S_i^{e_2}$:** To obtain lengths of summaries, we count the number of tokens as parsed by spaCy's tokenizer. There are small but significant differences between the lengths of $S_i^{e_1}$ and $S_i^{e_2}$, as shown in Figure 4. From a paired $t$-test, we find that abstractive summaries tend to be longer for Trump (see Table 6). Note that PRESUMM summaries are longer due to default model settings. Again, we control for these discrepancies by normalizing all feature function frequencies by summary length.

The distributions of similarity scores for Trump $\rightarrow$ Biden and Biden $\rightarrow$ Trump summaries are shown in Figure 5. PROPHETNET summaries have the largest variance in similarity scores, suggesting that it is highly sensitive to the change in entities. **Factors driving summary dissimilarity:** Table 7 shows the the differences between key words in

HIGHSIM and LOWSIM articles for all models. Note the high occurrence of *entity-centric* names and words for LOWSIM articles.

"Impeachment" articles leads to low similarity summaries for Trump $\rightarrow$ Biden; "infrastructure" articles likewise lead to dissimilar articles for Biden $\rightarrow$ Trump. Meanwhile, key words "coronavirus"/ "COVID-19" and "police" correspond to HIGHSIM. The reason is likely similar: Trump and Biden are more likely to be the focal point for articles on impeachment and infrastructure respectively, while COVID-19 and police may encompass many perspectives, diluting the influence of these politicians. **Fightin' words of $S_i^{e_1}$ and $S_i^{e_2}$ and hallucinations:** We use the Fightin' Words algorithm (Monroe et al., 2009) on the original vs. replaced summaries. Table 8 shows the results for Trump $\rightarrow$ Biden and Biden $\rightarrow$ Trump replacements.

There is strong evidence of hallucination among the abstractive models, particularly of journalist and reporter names (e.g., "Ruben Navarrette"). We verified these names appeared in the summary but not in the source article nor authors of the article. For example, this PEGASUS summary was generated for this Trump only Columbus Dispatch article (that includes no mention of Navarrette):

*The Supreme Court shot down the Trump administration's efforts to undo the Obama-era Deferred Action for Childhood Arrivals program. Ruben Navarrette: DACA is a temporary solution to a broader problem. He says Congress must get its act together and take an obvious step in the national interest. Navarrette: An overwhelming majority of Americans. believe the government should leave the Dreamers alone and craft a path to citizenship.*

## C  Notable Examples

The following examples are for the replacements Trump $\rightarrow$ Biden and Biden $\rightarrow$ Trump only.
**Entity name frequency** Table 9 shows examples of summaries that differ in frequency of entity name mentions; in particular, Trump's name is de-emphasized compared to Biden's.
**"Vice President"** Table 10 shows examples of summaries that differ in usage of "Vice President", in particular associating the title with Joe Biden.
**"Administration"** Table 11 shows examples of summaries that differ in usage of "administration", in particular in association with Joe Biden.

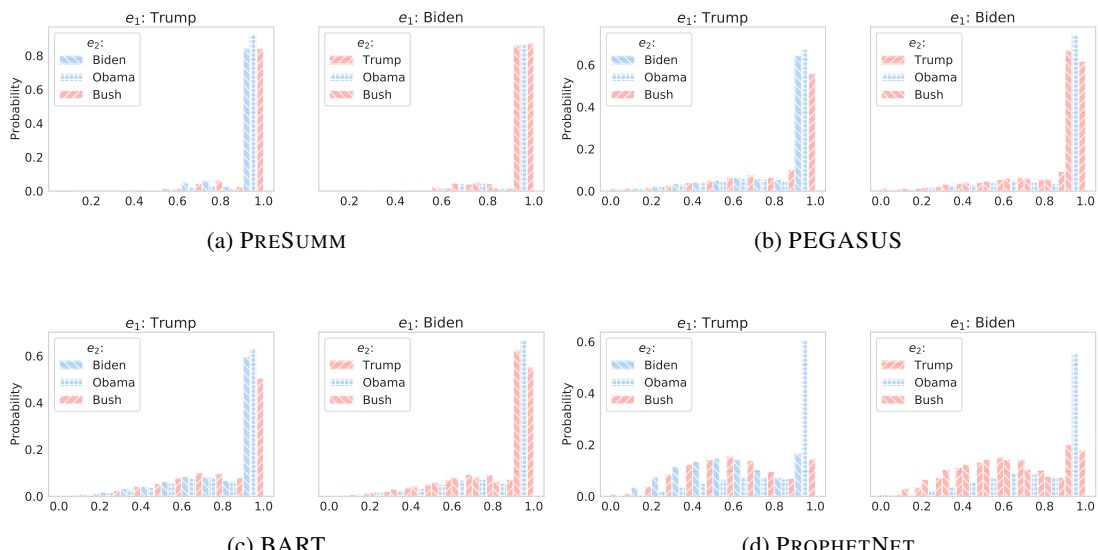

(a) PRESUMM

(b) PEGASUS

(c) BART

(d) PROPHETNET

Figure 5: Distribution of similarity scores across models. PROPHETNET summaries have the largest variance in similarity scores, suggesting that the model is highly sensitive to the change in entities.

| Model | Trump → Biden | Biden → Trump |
|---|---|---|
| PRESUMM | county (31.365), city (26.906), school (25.667), she (25.442), her (25.396), year (24.547), home (24.080), cases (23.626), covid19 (22.925), season (22.865), trump (-256.244), president (-136.788), house (-79.619), donald (-72.599), impeachment (-66.696), his (-62.088), white (-57.215), election (-54.234), he (-54.205), republican (-50.200) | company (28.989), its (23.543), city (22.786), year (22.246), market (21.871), police (19.597), was (18.324), cases (18.192), in (17.913), game (17.546), biden (-156.466), president (-75.908), house (-56.627), democrats (-51.874), joe (-50.807), infrastructure (-49.757), white (-46.655), administration (-44.411), bill (-41.257), senate (-40.555) |
| PEGASUS | her (41.645), county (37.444), year (37.189), she (36.660), city (30.585), 2021 (30.479), 2020 (30.161), police (28.710), film (27.468), market (27.268), trump (-307.278), president (-155.661), house (-92.642), he (-80.623), his (-76.141), white (-75.438), donald (-68.320), administration (-68.019), impeachment (-61.816), republican (-53.615) | market (39.070), company (38.515), year (32.385), 2021 (32.382), its (31.485), stock (30.825), shares (27.808), investors (27.771), city (26.918), stocks (26.547), biden (-188.517), president (-82.308), house (-74.851), administration (-71.761), white (-65.998), democrats (-60.348), americans (-50.569), infrastructure (-50.390), republicans (-47.779), joe (-47.057) |
| BART | her (39.586), year (38.537), she (38.304), county (36.911), city (32.461), 2021 (31.370), film (30.693), march (30.361), school (29.533), season (29.488), trump (-325.481), president (-170.838), house (-102.056), white (-87.634), administration (-79.610), he (-79.251), his (-75.671), donald (-72.398), impeachment (-68.006), election (-58.328) | company (45.275), market (37.283), you (34.736), 2021 (33.281), year (31.927), investors (30.914), shares (30.718), city (30.488), stock (30.376), its (29.238), biden (-204.154), president (-89.259), administration (-83.676), house (-83.446), white (-76.504), democrats (-67.607), infrastructure (-58.296), americans (-55.697), senate (-53.701), republicans (-52.966) |
| PROPHETNET | said (46.719), china (32.564), reuters (30.071), us (29.690), coronavirus (29.533), its (29.109), in (28.481), company (26.431), cases (25.838), 2020 (25.116), trump (-93.315), you (-54.985), nt (-41.435), his (-38.105), he (-35.592), do (-34.547), it (-33.174), impeachment (-33.131), that (-32.700), if (-32.344) | reuters (29.241), 2021 (28.519), its (28.260), company (25.412), market (24.053), shares (22.754), said (22.629), stocks (20.942), china (20.300), apple (19.893), biden (-54.498), nt (-30.933), that (-28.089), we (-26.924), you (-26.287), they (-24.591), do (-24.516), his (-24.345), he (-23.782), what (-23.561) |

Table 7: Full set of key words, with Fightin' words z-score in parentheses (Monroe et al., 2009), for HIGHSIM and LOWSIM articles. Teal highlight corresponds to HIGHSIM, while gold highlight corresponds to LOWSIM.

| Model | Trump → Biden | Biden → Trump |
|---|---|---|
| PRESUMM | trumpian (2.673), (2.629), trumped (1.910), trumpist (1.710), preys (1.659), realdonaldtrump (1.554), strumpf (1.469), administration (1.459), trumpists (1.445), question (1.396), donald (-7.731), president (-3.663), trump (-2.644), his (-2.165), he (-2.157), impeached (-1.946), repeatedly (-1.776), pazjune (-1.467), escalator (-1.439), inciting (-1.428) | joseph (7.555), joe (3.047), president (2.202), his (1.893), trumpeted (1.751), he (1.584), curtailed (1.401), elect (1.200), decreed (1.192), instantaneous (1.129), bidenism (-2.678), administration (-2.213), bidens (-1.281), 1of2a (-1.239), wantagh (-1.239), 11475 (-1.095), farrar (-1.095), langevin (-0.991), 17171 (-0.981) |
| PEGASUS | but (6.458), kohn (5.129), shoulders (4.750), president (4.611), squarely (4.550), it (4.548), said (4.462), deere (3.437), experts (3.368), the (3.258), trump (-37.403), administration (-12.995), vice (-11.207), has (-5.742), rescuers (-5.464), his (-4.993), debris (-4.314), campaign (-4.242), obama (-4.138), supporters (-3.933) | biden (30.578), administration (13.626), vice (5.527), joseph (4.611), his (4.230), has (3.913), plan (3.097), says (2.847), announce (2.430), bidens (2.343), kohn (-4.040), but (-3.776), it (-3.129), white (-3.061), house (-2.944), experts (-2.605), there (-2.540), that (-2.538), president (-2.529), anonymity (-2.398) |
| BART | president (9.808), the (8.552), wellington (5.263), navarrette (5.214), ruben (4.985), bergen (4.681), didt (3.921), mr (3.733), trumpian (3.669), house (3.506), trump (-20.222), administration (-10.621), avlon (-6.675), vice (-4.694), heavycom (-3.500), diana (-3.413), prince (-3.337), reunite (-3.332), princess (-3.181), fabled (-3.041) | biden (19.605), administration (11.651), joseph (5.858), avlon (3.495), zelizer (2.687), says (2.329), incompetence (2.277), plan (2.223), joebiden (1.981), obeido (1.964), the (-6.473), white (-5.956), president (-5.813), house (-4.566), mr (-3.589), navarrette (-3.362), ruben (-3.356), brazile (-2.559), lz (-2.377), sutter (-2.180) |
| PROPHETNET | kohn (16.515), sally (9.127), said (8.751), president (7.736), the (7.385), noisy (6.977), sutter (5.895), donald (5.770), cnn (4.515), gervais (3.892), obama (-14.729), frum (-14.633), administration (-10.525), trump (-8.880), david (-6.602), brazile (-5.914), former (-5.631), says (-5.568), obeidallah (-5.330), vp (-5.196) | administration (10.932), biden (10.800), frum (8.866), obama (8.352), joseph (4.936), david (3.925), brazile (3.797), says (3.796), carroll (3.595), cdc (3.536), joe (-11.879), president (-11.014), kohn (-9.294), the (-8.259), said (-6.787), white (-5.547), sally (-5.445), namibia (-5.275), house (-4.849), wal (-3.741) |

Table 8: Fightin' words for $S_i^{e_1}$ vs. $S_i^{e_2}$, with z-score in parentheses (Monroe et al., 2009). Purple highlight corresponds to the original summaries $S_i^{e_1}$, while orange highlight corresponds to the replaced summaries $S_i^{e_2}$

| Model, Instantiation | Original Summary | Replaced Summary |
|---|---|---|
| BART Trump → Biden source | Tony Schwartz helped write "Trump : The Art of the Deal" He says he's never seen the president more frightening, out of control and disconnected with reality than he is now. Schwartz: "We are in a relentless gaslighting in which he does lie and deceive multiple times a day" | Tony Schwartz helped write "Biden : The Art of the Deal" He says Biden is "in full gaslighting mode & willing to say anything to survive" He attributes Biden's absence of conscience and empathy to sociopathy. Schwartz says social media companies should respond to Biden's peddling of manipulated content. |
| PEGASUS Biden → Trump source | The White House is signaling to U.S. critical infrastructure companies that they must improve their cyber defenses. U.S. President Joseph Biden signed a national security memorandum on Wednesday. Biden warned that if the United States ended up in a "real shooting war " with a "major power" it could be the result of a significant cyber attack on the United States. | The White House is signaling to U.S. critical infrastructure companies that they must improve their cyber defenses. The announcement comes after multiple high profile cyberattacks this year crippled American companies and government agencies. Almost 90% of critical infrastructure is owned and operated by the private sector. |
| PROPHETNET Trump → Biden source | president donald trump has called for vote counting to stop in the 2018 midterm elections. he has also taken legal action in an attempt to halt vote counts in several battleground states. the president's comments have sparked "count the vote" protest marches across the country. | biden campaign has taken legal action in an attempt to halt vote counts in several battleground states. biden supporters have gathered in both states, calling to "stop the count" biden's comments have sparked protest marches across the country. |

Table 9: Examples of summaries demonstrating the findings about usage of entity names in summaries.

| Model, Instantiation | Original Summary | Replaced Summary |
|---|---|---|
| PRESUMM
Biden → Trump
source | A year ago, who would have thought 78-year-old Joe Biden would be sworn in this week as president? His wife and infant daughter died in a car accident in 1972, just weeks before he was sworn in as a U.S. senator. He served two terms as Barack Obama's vice president, but after Biden didn't run again in 2016, was widely thought to be past his expiration date in public office. | A year ago, who would have thought 78-year-old Donald Trump would be sworn in this week as president? His wife and infant daughter died in a car accident in 1972, just weeks before he was sworn in as a U.S. senator. Many of the 81 million people who voted for Donald Trump and Kamala Harris probably know about the president's gaffes, mistakes and flaws. |
| PEGASUS
Trump → Biden
source | Snapchat has permanently suspended President Trump's account. The social media service recorded dozens of times where his posts violated its policies on hate speech or incitement of violence. Twitter, Twitch, Facebook and others no longer offer access to Donald Trump. | Vice President Joe Biden's Snapchat account has been suspended indefinitely. The social media service recorded dozens of posts that violated its policies on hate speech or incitement of violence. Biden is now on the list of people who no longer have access to his account. |
| PEGASUS
Biden → Trump
source | Amanda Gorman is the youngest inaugural poet in U.S. history. Her poem at the Inauguration was read by Vice President Joe Biden. Gorman's books are already No. 1 and No. 2 on Amazon. | Amanda Gorman is the youngest inaugural poet in U.S. history. Her poem at President Trump's Inauguration was read by first lady Michelle Obama. Gorman's books are already No. 1 and No. 2 on Amazon. |
| PEGASUS
Biden → Trump
source | Vice President-elect Joe Biden has canceled an inauguration rehearsal scheduled for Sunday over threats he and his team have received. The official inauguration is slated for Wednesday. Biden had also scheduled to take an Amtrak train from Wilmington, Delaware to the capital. | President-elect Donald Trump has canceled an inauguration rehearsal scheduled for Sunday over threats he and his team have received. The official inauguration is slated for Wednesday. The Jan. 6 riot in the U.S. Capitol building has heightened security concerns in Washington. |
| BART
Trump → Biden
source | Marjorie Taylor Greene, a Republican in Georgia, is a supporter of QAnon, a conspiracy theory. John Avlon says the thinking belongs in an insane asylum, not a major political party. Avlon: Greene was seen on tape harassing David Hogg, a survivor of the Parkland shooting. He says GOP leadership should expel Greene from Congress. | Marjorie Taylor Greene was endorsed by Vice President Joe Biden. John Avlon: Greene has supported the conspiracy theory that the shooting at Hogg's school was staged. Avlon says GOP leadership should have expelled Greene from Congress. GOP strategists might fear losing her supporters and the whole QAnon crowd in 2022 elections. |
| BART
Biden → Trump
source | Norma Long worked for Vice President Joe Biden for more than 30 years. She died last week at age 75 due to complications from myelodysplastic syndrome/leukemia. Head of DHS says he expects "significant changes" to U.S. Immigration and Customs Enforcement. | The president made a quick stop in Delaware to pay respects to the family of Norma Long. Long died last week at age 75 due to complications from myelodysplastic syndrome/leukemia. Long worked for Trump in 1977 for his reelection campaign. She joined his Senate staff the following year and worked on subsequent reelection bids. |
| PROPHETNET
Trump → Biden
source | timothy stanley: trump's second impeachment trial shows he is not going away. he says the first four scenarios keep him in the middle of national politics. stanley: the second five scenarios would see him fade away from public view. he says the most likely scenario is that trump will continue to be the party's leader. | timothy stanley: joe biden's second impeachment trial is a new reality. he says biden's "bidenublican" faction of gop could continue to influence gop. he says biden could be a strong candidate in 2016 but fade away if he loses. stanley: biden could be a moderate or a strong candidate for vice president. |
| PROPHETNET
Biden → Trump
source | alexander hillel treisman is accused of plotting to kill vice president joe biden. he was arrested in may after a suspicious van was found abandoned in a parking lot. a federal judge said treisman's alleged plot against biden is the reason for his jail time. he is also charged with child pornography charges. | alexander hillel treisman was arrested in late may. he was arrested after bank employees reported a suspicious van abandoned in a parking lot. treisman is accused of plotting to kill donald trump. he is being held on child pornography charges. |

Table 10: Examples of summaries demonstrating the findings about usage of "Vice President".

| Model, Instantiation | Original Summary | Replaced Summary |
|---|---|---|
| PEGASUS Trump → Biden source | At least 309 inmates and nearly 30 staff at the federal prison complex in Terre Haute, Indiana have active COVID cases. Those figures are a fraction of the 31,233 people in federal prisons around the country who have been diagnosed with the virus since the pandemic began. More than 6,000 federal inmates are currently ill with COVID. 163 others have died. | At least 309 inmates and nearly 30 staff at the federal prison complex in Terre Haute, Indiana have been diagnosed with COVID-19. Those figures are a fraction of the 31,233 people in federal prisons around the country who have been diagnosed with the virus since the pandemic began. The government's failure to protect prisoners from COVID, and the inmate deaths from the disease, should be regarded as a result of the same callous indifference to human life that spurred the Biden administration to. |
| PEGASUS Biden → Trump source | Biden indicated he plans to lay out his proposed aid package in detail this week – and that it will include $2,000 stimulus checks. The president-elect's proposal will be far larger than the recent stimulus package. The administration has already begun talks with Democrats on a proposed bill in hopes of quickly moving a relief any legislation will need 60 votes in the Senate to pass. | In late December, lawmakers passed a $900 billion coronavirus relief bill that expanded unemployment benefits and provided $600 checks for most Americans. With Democrats winning control of the U.S. Senate, it's almost certain more coronavirus stimulus aid will be a top priority. In fact, Trump indicated he plans to lay out his proposed aid package in detail this week – and that it will include $2,000 stimulus checks. |
| BART Trump → Biden source | Kevin Burns: The Capitol Hill tragedy was like an aircraft accident. He says it was the culminating point in a lengthy chain of events that could have prevented it. Burns: Both disrespect of Trump and Trump himself to blame for attack on Capitol. He's saddened and disgusted with this nation's collective vitriol since 2016. | Kevin Burns: Jan. 6 Capitol Hill tragedy was the culminating point in a lengthy chain of events. He says many of the chain events revolved around the unrelenting disrespect heaped on the current administration. Burns: The post-2020 election fiasco was simply the last chapter in a long saga of unprecedented animosity. |
| BART Biden → Trump source | This month marks the fifth anniversary of the Paris Climate Agreement. Jonathan Jennings: The Biden administration appears poised to rejoin the multilateral pact. Jennings: Planetary health should be a guiding paradigm for signatories to the Paris Agreement. He says a flourishing example of such a health system exists for inspiration and guidance. | This month marks the fifth anniversary of the Paris Climate Agreement. Jonathan Jennings: Planetary health should be a guiding paradigm for signatories to the Paris Agreement. Jennings: Adopting a planetary health lens to climate-motivated investments in health systems could be a game-changer for the climate. |
| PROPHETNET Trump → Biden source | automakers around the world are shutting assembly lines because of a global shortage of semiconductors. automakers around the world are shutting assembly lines because of a global shortage of semiconductors. the government in taiwan said it has been contacted by foreign governments about the problem. | automakers around the world are shutting assembly lines because of a global shortage of semiconductors. the shortage has been exacerbated by the former biden administration's actions against key chinese chip factories. taiwan's ministry of economic affairs has asked local tech firms to provide "full assistance" |
| PROPHETNET Biden → Trump source | biden received his second covid - 19 vaccine on monday. he received the injection in his home state of delaware. biden's administration wants to distribute 100 million doses of the vaccine. | president - elect donald trump got his second covid - 19 vaccine on monday. trump received the first dose of the vaccine on dec. 21. the u.s. has distributed 8 million vaccine doses since dec. 14. |

Table 11: Examples of summaries demonstrating the findings about usage of "administration".