# OpenReview forum: "Entity-Based Evaluation of Political Bias in Automatic Summarization"
_EMNLP/2023/Conference — EMNLP 2023 Findings_

### Official Review · Reviewer_f6t5 · 2023-07-21

**Soundness:** 4

**Excitement:**

4: Strong: This paper deepens the understanding of some phenomenon or lowers the barriers to an existing research direction.

**Justification For Ethical Concerns:**

Although bias is in general a research topic sensitive to ethical concerns, the context of this paper seems ok.

**Missing References:**

- Liu, R., Jia, C., Wei, J., Xu, G., & Vosoughi, S. (2022). Quantifying and alleviating political bias in language models. Artificial Intelligence, 304, 103654.
- Nayeon Lee, Yejin Bang, Tiezheng Yu, Andrea Madotto, and Pascale Fung. 2022. NeuS: Neutral Multi-News Summarization for Mitigating Framing Bias https://aclanthology.org/2022.naacl-main.228. In _Proceedings of the 2022 Conference of the North American Chapter of the Association for Computational Linguistics: Human Language Technologies_, pages 3131–3148, Seattle, United States. Association for Computational Linguistics.

**Paper Topic And Main Contributions:**

The authors investigate political bias in summarization models by using entity replacement of Donald Trump and Joe Biden. They find consistent differences in resulting summaries after replacement compared to before replacement. The results imply that political bias is encoded in the LM.

**Questions For The Authors:**


- It may be interesting to do a bit more comparison to types of bias such as framing bias that other papers have used in relation to politics and summarization.
- gpt3 / current lms not included, but they refer to that in their limitations section as well.

As more future research questions
- : LMs are in essention aimed at encoding some sort of bias as they encode likeliness of words following each other. This means that it's also logical that it has certain associations with certain entities based on other data. By replacing entities you place one entity in the context of another. This is asking the LM to go against the previously encoded associations (particularly with such two divided parties as Biden and Trump). How realistic is it to expect that the LM can do that bias-free based on one article only?
- And to what extend is it political bias of the LM model we are now speaking of, or the bias of the context of the articles? Aka in general, how often do journalists associate Trump with an administration - is the difference in the LMs because of the LMs or because of data the LMs was trained on?

**Reasons To Accept:**

- methodology of keeping everything the same except the entity name is simple and strong
- code and data will be released
- has important consequences for example related to how quickly (or not quickly) associations change in a language model (aka president vs vice president )
- good awareness of limitiations

**Reasons To Reject:**

- by only comparing Trump, Biden, Bush, Obama with such strong presences it is limited into the generalizability, but it shows at least the potential of the method.
- a lot of important information for being able to assess the results yourself are now put in the appendices. Particularly appendix A and B

**Reproducibility:**

3: Could reproduce the results with some difficulty. The settings of parameters are underspecified or subjectively determined; the training/evaluation data are not widely available.

**Reviewer Confidence:**

3: Pretty sure, but there's a chance I missed something. Although I have a good feel for this area in general, I did not carefully check the paper's details, e.g., the math, experimental design, or novelty.

**Typos Grammar Style And Presentation Improvements:**

Tables. Although the arrows are explained in table 1, I find the tables hard to read. Particularly because they represent changes to 'something' and there are multiple possibilities there. I prefer the tables to be readable by themselves so that the table description contains relevant info. Also I would prefer to see the actual means to give an impression into what range these frequencies fall. This would also help in reproducing/checking results.

The authors mention that their framework can be used for further analysis of bias in summarization models, but to me the focus of the paper was more on the results than on the framework (they were also combined in one section) - it may strengthen the paper if the framework stands out more from the actual results.

Now there are 4 models compared, but the differences between the 3 abstractive models are seemingly small and are also not really compared to each other. To save space you could also just use one of the 3 abstractive models to make your point.

---

> ### Author Rebuttal · Authors · 2023-08-28
>
> Thank you for the encouraging review and comments.
>
>
> __Re: limited generalizability__
> We acknowledge that experimenting with additional pairs of entities will support our findings more robustly. Nonetheless, because Trump and Biden are prominent political figures that are featured heavily in US political news (L184-5), the consistent biases we found across multiple summarization models with the large amount of data still constitute important initial findings.
>
>
> __Re: results in appendices__
> We unfortunately had to move these appendix sections from the main paper due to space constraints. We did our best to address these methods and results concisely in the main paper. Are there specific parts of these appendix sections you think should be included in the main paper? If accepted, we will have an extra page for the changes you suggest.
>
>
> __Re: comparing to framing bias__
> There is strong resemblance between selecting the most important information in the summarization process and the phenomenon of media framing, where media outlets selectively report aspects of an issue or quotes of politicians (Entman 1993, Chong and Druckman 2007). A relevant example is the framing of “Obamacare”' vs. the Affordable Care Act (Hopper 2015). The framing bias/selection process by media outlets has received substantial attention in computational research (Niculae et al. 2015, Tan et al. 2018, Guo et al. 2022, etc.). Some differences with our set-up is that the selection is done by models rather than humans/media outlets and that the kind of bias may not align with the conservative vs liberal biases as defined for media framing.
>
>
> __Re: bias of training data/article context vs model__
> We agree that the differences we uncover may have arisen from biases in representation from pre-training data, and the finding that biases do exist in the models is not necessarily surprising. These are widely used pre-training datasets, so these behaviors, wanted or not, may extend to even more models. Regardless, it is not always desirable for models to completely parrot the biases of their training data. Furthermore, as described in L262, politicians are often in similar situations, such as giving a speech or meeting a world leader, and producing systematically different summaries of an event solely based on the entity participating may exacerbate misinformation, polarization, and other downstream effects.
>
> We focus on exposing these differences specifically for automatic summarization models for the task of news summarization. We are not arguing that summarization models should be bias-free; in fact, nuanced representations of entities indicate capable models. Rather we are trying to stimulate the discussion of what summarization model behaviors are ultimately desirable with respect to political entities.
>
> __Re: additional references and clarity suggestions__
> Thank you for these helpful suggestions! We apologize for the confusing arrow notation; it has been used in other ACL papers, e.g., [Zong et al.](https://aclanthology.org/2020.acl-main.473/), [Tan et al]( https://aclanthology.org/P14-1017/), but we will revise our captions to be more self-contained and accessible. We will also include the numerical means for reproducibility.
>
> While the results in the main paper are mostly the same for the 3 abstractive models, we believe it is still useful to show all their differences, especially since they have different pre-training objectives. Specific distinctions between the abstractive models are shared in the appendix, i.e., examples of hallucination in Table 8.  Furthermore, ProphetNet (the “latest” model compared in this study) summaries have the largest variance in similarity scores, suggesting that the model is highly sensitive to the change in entities (Figure 5).

---

### Official Review · Reviewer_utt1 · 2023-08-04

**Soundness:** 3

**Excitement:**

3: Ambivalent: It has merits (e.g., it reports state-of-the-art results, the idea is nice), but there are key weaknesses (e.g., it describes incremental work), and it can significantly benefit from another round of revision. However, I won't object to accepting it if my co-reviewers champion it.

**Paper Topic And Main Contributions:**

The paper investigates the political bias present in automatic summarization models and aims to shed light on the portrayal of politicians in generated summaries. The authors use an entity replacement method to examine the differences in summaries of news articles when key political figures, Donald Trump and Joe Biden, are replaced with each other. The study demonstrates consistent variations in summaries, suggesting different model representations of the two politicians. The paper highlights the importance of understanding bias in summarization models and emphasizes the need for responsible deployment and usage.

**Questions For The Authors:**

Question A: When conducting the paired t-tests to measure significance levels, did you consider controlling for any confounding variables or potential biases that could influence the results?

Question B: You mentioned that entity-centric articles led to more dissimilar summaries upon entity replacement. Could you provide some insights into why entity-centric articles might result in more dissimilar summaries and how this observation aligns with your research objectives?

**Reasons To Accept:**

The study provides empirical evidence of substantial differences across four summarization models when analyzing the summaries of news articles involving different politicians.
The research lays the groundwork for further exploration of biases in summarization models, encouraging future research and discussion in this essential research area.
The authors acknowledge the importance of considering normative questions related to what constitutes an ideal summary. They emphasize the need for responsible use of summarization models and provide a framework for assessing biases.

**Reasons To Reject:**

While the paper states that the term "bias" is used to refer to significant differences, it might be beneficial to clarify and define the specific type of bias being explored, to avoid misunderstandings or misinterpretations.

To strengthen the study's findings, the paper could include a comparative analysis with human-generated summaries to ascertain the extent of bias in automatic summarization models compared to human summarizers.

**Reproducibility:**

4: Could mostly reproduce the results, but there may be some variation because of sample variance or minor variations in their interpretation of the protocol or method.

**Reviewer Confidence:**

4: Quite sure. I tried to check the important points carefully. It's unlikely, though conceivable, that I missed something that should affect my ratings.

---

> ### Author Rebuttal · Authors · 2023-08-28
>
> Thank you for the careful review and questions.
>
> __Re: clarify bias__
> We acknowledge the term “bias” is overloaded and its usage can be confusing. We will emphasize that we are studying representational biases (differences) of political entities in automatic summaries.
>
> __Re: comparison with human-generated summaries__
> Our main goal is to study the bias in automatic summaries. Comparing that bias with human bias could be interesting, but it does not change the normative nature of the question. Additionally humans can also produce “biased” summaries, so the normative question of what constitutes an ideal summary is still unanswered. Indeed, our conclusions would challenge the notion that human summaries should be considered neutral or gold standard.
>
> __Re: control for confounding variables__
> Summary length is the main confounding variable, which we controlled for by normalizing the frequencies before conducting the t-tests. Since the source article context was controlled for and model configurations were the same between summaries, other differences seem likely to stem from the replaced entity’s effects on that particular model.
>
> __Re: entity-centric insights__
> Yes! In Appendix B (L577-601), we elaborate on the entity-centric insights – these were omitted from the main paper for space. We found that articles with high frequencies of the original entity’s name lead to more dissimilar summaries upon replacement. This supports our finding that the summarization models have different representations for these entities, since the greater focus on the entity in the article results in more variations between the original vs. replaced summaries.
>
> Furthermore, articles featuring more key words like “impeachment” and “infrastructure” also led to greater dissimilarity, while words like “police” and “COVID-19” did not result in as much difference upon replacement. This is likely due to Biden and Trump being the focal point for articles on infrastructure (Biden’s infrastructure bill) and impeachment (of Trump) respectively, while COVID-19 and police news may encompass many perspectives and dilute the influence of these politicians. Overall, these insights make it even more important to consider desirable aspects of automatic summaries, since such sensitivity to political entities can result in summaries with unintended effects on readers.

---

### Official Review · Reviewer_7A5T · 2023-08-04

**Soundness:** 3

**Excitement:**

3: Ambivalent: It has merits (e.g., it reports state-of-the-art results, the idea is nice), but there are key weaknesses (e.g., it describes incremental work), and it can significantly benefit from another round of revision. However, I won't object to accepting it if my co-reviewers champion it.

**Paper Topic And Main Contributions:**

The paper introduces a new approach of employing entity replacement to analyze the portrayal of politicians in automatically generated summaries. By comparing summaries with and without specific political entities, the authors conduct experiments to show consistent differences in how Trump vs Biden are represented in ~200,000 news articles, indicating that potential reporting biases present in summarization models.

**Reasons To Accept:**

The use of an entity replacement method to analyze the portrayal of politicians in automatically generated summaries is interesting and has intellectual merit.

The observation and conclusion it leads to, i.e. "summarization model biases towards learn different representations for different entities and provide consistently different summaries depending on the  entities involved.", presents an important and interesting research problem for the community to address.

**Reasons To Reject:**

In the paper's current form, there doesn't seem to be an definitive conclusion which will help future researchers understand "what exactly we need to work on to improve summarization models/task definitions". IMO there are many open questions left hanging that could be answered with further experiments and analysis, which will greatly strengthen the overall presentation + claims made in the paper.

Here are some improvements that I would suggest:
1. With all summarization models, include what summarziation datasets they are trained on, as this directly governs model's behavior on what types of summary it will produce. It would be much easier to read + understand your results that way.
2. Include analysis for more "control" variables, e.g. does the source of a document matter for what kind of summaries models will produce? e.g. Do models tend to generate Trump less frequently than Biden in left-leaning vs. right-leaning sources? Including analysis like this will strengthen your claim that the bias comes from "summarization model", not something else.

Here are some potential extensions + further questions for analysis that the authors can think about --
1. What might be the cause of the Trump vs. Biden bias from the model or summarization task perspective. Again, from my intuition, the choice of training data for summarziation models will have a large impact on the results,  e.g. most of the models mentioned in the paper are trained on data from more left-leaning news sources.
2. Generalize the analysis to include more entities -- This will potentially reveal what types of entities do model exhibit biases towards, and help us understand the systematic causes behind the phenomena.

**Reproducibility:**

3: Could reproduce the results with some difficulty. The settings of parameters are underspecified or subjectively determined; the training/evaluation data are not widely available.

**Reviewer Confidence:**

4: Quite sure. I tried to check the important points carefully. It's unlikely, though conceivable, that I missed something that should affect my ratings.

---

> ### Author Rebuttal · Authors · 2023-08-28
>
> Thank you for the thorough review and questions.
>
> __Re: no definitive conclusion__
> The intended conclusions of this paper are (1) we need to recognize that there is no neutral summary and (2) the research community should revisit the definition of the summarization task. We emphasize that we are the first paper to concretely show that summarization models do not generate “neutral” summaries and that the notion of neutral summaries is flawed – existing literature and evaluation metrics of automatic summaries do not address neutrality/assume neutrality as a given, which we have demonstrated should not be the case. We agree there are many open questions inspired by our findings, but we believe these would be in the scope of future work. We hope that our short paper will start a discussion on these hard issues, and we respectfully ask the reviewer to evaluate this paper as a *short paper*.
>
>
> __Re: summarization models’ training data__
> In general, all models were trained on widely used pre-training datasets. These are the specifics for the models we used.
>
> - PreSumm (2019)
>     - training data: same as BERT
> - PEGASUS (2020)
>     -  training data: like T5, pre-trained on a very large corpus of web-crawled documents (C4, about ~46x bigger than BERT data)
>     - fine-tuned on CNN/DailyMail
> - BART (2019)
>     - training data:  same as RoBERTa (~10x bigger than BERT data)
>     - fine-tuned on CNN/DailyMail
> - ProphetNet (2020)
>     - training data:  same as RoBERTa (~10x bigger than BERT data)
>     - fine-tuned on CNN/DailyMail
>
> While there are differences in the exact training datasets used, all models’ training data ended before 2020, which is when our articles start. Additionally, all abstractive models used were fine-tuned on CNN/DM.
>
> We agree that Trump and Biden are popular figures, highly represented in the models’ pretraining data. The biases we uncover absolutely may have arisen from this data, and the finding that biases do exist is not necessarily surprising after the fact. Nonetheless, we believe it is still important to see what biases automatic summaries manifest; as such, we are the first to document political biases in the summarization setting. Investigating how these biases originate from the model training data is beyond the scope of this paper and can build off our initial findings. Furthermore, we emphasize that there are always “biases'' and we should rethink the qualities of ideal summaries rather than seeking neutral summaries.
>
> __Re: more control variables__
> These are great suggestions. Since our dataset (NOW corpus) includes all US news articles from a two year range, our findings are at least conclusive for the overall distribution of news.  As for differences between liberal vs. conservative news, we had preliminary results that showed no significant difference between media source leanings, which is why we did not include these in the paper. We will add this in the appendix in the revision or in the main paper if space allows.
>
>
> __Re: limited generalizability__
> We acknowledge that experimenting with additional pairs of entities will support our findings more robustly. Nonetheless, because Trump and Biden are prominent political figures that are featured heavily in US political news (L184-5), the consistent biases we found across multiple summarization models with the large amount of data still constitute important initial findings.
>
> In conclusion, we believe this paper contributes sound experiments and findings to the study of political bias in summarization models – a well-motivated social and technical problem. It invites the research community to work on this important yet understudied research area, and in particular, to consider the qualities of “ideal” automatic summaries.

---

### Meta-Review · Area_Chair_7ZXa · 2023-09-22

**Recommendation:** 4

**Metareview:**

The authors replace political entities in a given article/document and check their impact on summarization. There exists a significant political bias in summarization models.

Reviewers agree the task is interesting and the results also support the claim. This provides good directions for future research on the exploration of biases in summarization models (Reviewers utt1, f6t5).

However, results are restricted to some specific entities and carry geographical bias. There might be issues with the generalizability of the findings (Reviewer f6t5, 7A5T).

---

### Decision · Program_Chairs · 2023-10-07

**Decision:**

Accept-Findings

**Comment:**

The authors replace political entities in a given article/document and check their impact on summarization. There exists a significant political bias in summarization models.

Reviewers agree the task is interesting and the results also support the claim. This provides good directions for future research on the exploration of biases in summarization models (Reviewers utt1, f6t5).

However, results are restricted to some specific entities and carry geographical bias. There might be issues with the generalizability of the findings (Reviewer f6t5, 7A5T).